# Metabolites from the Paracel Islands Soft Coral *Sinularia* cf. *molesta*

**DOI:** 10.3390/md16120517

**Published:** 2018-12-19

**Authors:** Mei-Jun Chu, Xu-Li Tang, Xiao Han, Tao Li, Xiang-Chao Luo, Ming-Ming Jiang, Leen van Ofwegen, Lian-Zhong Luo, Gang Zhang, Ping-Lin Li, Guo-Qiang Li

**Affiliations:** 1Key Laboratory of Marine Drugs, Chinese Ministry of Education, School of Medicine and Pharmacy, Ocean University of China, Qingdao 266003, China; chumjun@163.com (M.-J.C.); 18765422150@163.com (X.H.); bruce13@163.com (T.L.); luoxc981@163.com (X.-C.L.); jiangmingming1989@126.com (M.-M.J.); 2Tobacco Research Institute, Chinese Academy of Agricultural Sciences, Qingdao 266101, China; 3College of Chemistry and Chemical Engineering, Ocean University of China, Qingdao 266100, China; tangxuli@ouc.edu.cn; 4Laboratory of Marine Drugs and Biological Products, National Laboratory for Marine Science and Technology, Qingdao 266235, China; 5Nationaal Natuurhistorisch Museum, 2300 RA Leiden, The Netherlands; ofwegen@yahoo.com; 6Xiamen Key Laboratory of Marine Medicinal Natural Products Resources, Xiamen Medical College, Xiamen 361023, China; lzluo@xmu.edu.cn (L.-Z.L.); zg@xmmc.edu.cn (G.Z.)

**Keywords:** Marine soft coral, *Sinularia*, terpenoids

## Abstract

Five new oxygenated sesquiterpenes, molestins A–D (**1**, **3**–**5**) and *epi*-gibberodione (**2**), three new cyclopentenone derivatives, *ent*-sinulolides C, D, and F ((+)-**9**–(+)-**11**), one new butenolide derivative, *ent*-sinulolide H ((+)-**13**), and one new cembranolide, molestin E (**14**), together with 14 known related metabolites (**6**–**8**, (–)-**9**–(–)-**11**, (±)-**12**, (–)-**13**, **15**–**19**) were isolated from the Paracel Islands soft coral *Sinularia* cf. *molesta*. The structures and absolute configurations were elucidated based on comprehensive spectroscopic analyses, quantum chemical calculations, and comparison with the literature data. Compound **5** is the first example of a norsesquiterpene with a de-isopropyl guaiane skeleton isolated from the genus *Sinularia*. Molestin E (**14**) exhibited cytotoxicities against HeLa and HCT-116 cell lines with IC_50_ values of 5.26 and 8.37 μM, respectively. Compounds **4**, **5**, and **8** showed significant inhibitory activities against protein tyrosine phosphatase 1B (PTP1B) with IC_50_ values of 218, 344, and 1.24 μM, respectively.

## 1. Introduction

Soft corals of the genus *Sinularia* (phylum Cnidaria, class Anthozoa, subclass Octocorallia, order Alcyonacea, suborder Alcyoniina, family Alcyoniidae), comprising more than 100 species, constitute an important invertebrate group occurring widely in different coral reefs of the world. The genus *Sinularia* is well known for being a rich source of valuable secondary metabolites as they have produced many structurally unique and biologically active compounds. Since 1975 [1,2], over 50 species of the genus *Sinularia* have been chemically investigated, showing a wide range of structure diversities in diterpenes (especially cembrane diterpenes) [3,4], sesquiterpenes [5,6], polyhydroxylated steroids [7,8], polyamines [9,10], cyclopentenones [6,8,11,12], and butenolides [11,12]. Many of these metabolites exhibited a wide range of biological activities, including cytotoxic [13,14], anti-inflammatory [15], antimicrobial [16], antifouling [17], and antifeedant activities [18]. 

Up to now, there have been few reports on the soft coral *S.* cf. *molesta* and *S. molesta*, except for one report on the Moyli Island *S. intacta* (a synonym of *S. molesta*) in 1999, resulting in a series of africane-type sesquiterpenes [19]. In addition, our first and preliminary study on *S.* cf. *molesta* yielded some terpenoids and sterols [20]. In our continuing interest in bioactive natural products from Paracel Islands invertebrates [8,21,22], we investigated extracts of the Paracel Islands soft coral *S.* cf. *molesta* that exhibited cytotoxic activities against human leukemia cell lines (K562) and human myeloid leukemia cell lines (HL-60), with the inhibition ratios of 60.0% and 72.8%, respectively, at the concentration of 50 μM. That led to the isolation of two new secoguaiane-type sesquiterpenes (**1** and **2**), three new guaiane-type sesquiterpenes (**3**–**5**), three new cyclopentenone derivatives ((+)-**9**–(+)-**11**), one new butenolide derivative ((+)-**13**), and one new furanocembranolide (**14**), together with 14 known analogs (**6**–**8**, (–)-**9**–(–)-**11**, (±)-**12**, (–)-**13**, and **15**–**19**) (Figure 1). Their cytotoxic and inhibitory activities against protein tyrosine phosphatase 1B (PTP1B) were evaluated. Herein, we report the isolation, structure elucidation, and bioactivities of these compounds.

## 2. Results and Discussion

Molestin A (**1**) was obtained as a colorless oil. Its molecular formula of C_15_H_24_O_3_ was established through a protonated molecular [M + H]^+^ at *m*/*z* 253.1803 (calcd for C_15_H_25_O_3_, 253.1798) in the (+)-high-resolution electrospray ionization mass spectrometry (HRESIMS) spectrum, indicating four degrees of unsaturation. The infrared radiation (IR) spectrum displayed absorption bands for hydroxy (3151 cm^−1^), carbonyl (1712 cm^−1^), and olefinic (1649 cm^−1^) functionalities. The ^13^C NMR and DEPT data (Table 1) of **1** exhibited 15 carbon resonances corresponding to four methyls, four methylenes, three methines, and four quaternary carbons (including one oxygenated, one olefinic, and two carbonyl). The olefinic carbons at *δ*_C_ 168.7 (C-7) and 127.1 (C-6) and the carbonyl carbon at *δ*_C_ 203.9 (C-5) showed characteristic chemical shifts for an *α*,*β*-unsaturated ketone (Table 1). The presence of a 3-oxobutyl group was readily verified by ^1^H–^1^H COSY correlations of H-1/H_2_-2/H_2_-3 and HMBC correlations from H_3_-11 to C-3 and C-4 (Figure 2). Furthermore, ^1^H–^1^H correlation spectroscopy (^1^H–^1^H COSY) correlations of H_2_-8/H_2_-9/H-10/H-1 and heteronuclear multiple-bond correlation (HMBC) correlations from H-6 to C-1 and C-8, and from H_2_-9 to C-1 and C-7 suggested that compound **1** should possess a conjugated cycloheptenone core with a 3-oxobutyl side chain linked to C-1. Additionally, a doublet methyl at *δ*_H_ 0.80 (3H, H_3_-15) was connected to C-10 by ^1^H–^1^H COSY correlation of H-10/H_3_-15 and HMBC correlations from H_3_-15 to C-1, C-9, and C-10. Besides, signals of two downfield singlet methyls at *δ*_H_ 1.38 (6H, H_3_-13 and H_3_-14) and an additional oxygenated quaternary carbon at *δ*_C_ 74.0 (C-12), in combination with the molecular formula of **1**, indicated the presence of a 2-hydroxyprop-2-yl group, which was attached to C-7 by HMBC correlations from H_3_-13 and H_3_-14 to C-7, and from H-6 and H_2_-8 to C-12. Accordingly, compound **1** was thus elucidated as the 12-hydroxylated derivative of the co-isolated secoguaiane-type sesquiterpene **6** (gibberodione), which was previously isolated from a Formosan soft coral *S. gibberosa* [5]. 

The absolute configuration of **1** was determined to be the same as that of gibberodione, according to the nuclear overhauser effect spectroscopy (NOESY) experiment, comparison of corresponding chemical shifts and optical rotations with those reported analogs, and electronic circular dichroism (ECD) calculations. The *E*-geometry of the Δ^6(7)^-double bond was assigned by NOESY correlations of H-6/H_3_-13 and H-6/H_3_-14 (Figure 3). NOESY correlations of H-1/H-10 and H_2_-2/H_3_-15 as well as the absence of H-1/H_3_-15 indicated the *cis*-orientation of H-1 and H-10, which was further supported by the chemical shifts of H_3_-15 at *δ*_H_ 0.80 and *δ*_C_ 16.3 recorded in CDCl_3_ (Ahmed et al. concluded that the chemical shifts of the secondary methyl at *δ*_H_ 0.83 and *δ*_C_ 15.7 in (1*S*, 10*S*) isomer were upfield, compared to *δ*_H_ 1.09 and *δ*_C_ 19.9 in (1*R*, 10*S*) isomer) [5]). The absolute configuration of **1** was determined by ECD calculations performed by the time dependent density functional theory (TDDFT)/ECD method [23]. The experimental ECD spectrum of **1**, which matched well with the calculated ECD spectrum for (1*S*, 10*S*) isomer, and showed mirror-image-like relationship with calculated ECD spectra for (1*R*, 10*R*) isomer (Figure 4), proved the (1*S*, 10*S*) absolute configuration for **1**, corresponding to the same sign of optical rotation of **1** ([*α*${]}_{D}^{20}$ +10. 0, *c* 0.1, MeOH) as that of gibberodione ([*α*${]}_{D}^{20}$ +20.8, *c* 0.72, CHCl_3_).

*Epi*-gibberodione (**2**), afforded as a colorless oil, was assigned a molecular formula of C_15_H_24_O_2_ by (+)-HRESIMS ion at *m*/*z* 237.1852 [M + H]^+^ (calcd for C_15_H_25_O_2_, 237.1849) and ^13^C NMR data (Table 1). 1D NMR (Table 1 and Table 2) together with ^1^H–^1^H COSY data of **2** revealed the presence of an *α*,*β*-unsaturated ketone, a 3-oxobutyl moiety, and an isopropyl group, implying compound **2** could be an analog of gibberodione [5]. That speculation was subsequently confirmed by detailed analyses of the ^1^H–^1^H COSY, heteronuclear singular quantum correlation (HSQC), and HMBC spectra, which established that compound **2** possessed an identical planar structure as gibberodione. NOESY correlations of H-6/H_3_-13 and H-6/H_3_-14 defined *E*-geometry of the Δ^6(7)^-double bond. Besides, NOESY correlations of H-1/H_3_-15 and the absence of H-1/H-10 indicated the *trans*-orientation of H-1 and H-10, which were confirmed by the chemical shift of H_3_-15 at *δ*_H_ 1.10 recorded in CDCl_3_ (see Section 3.3), because of the previous report that the *cis*-orientation isomer showed marked upfield shift for H_3_-15 relative to that of *trans*-orientation isomer and the upfield shift of H_3_-15 in gibberodione (*δ*_H_ 0.79) was detected [5]. The overall pattern of experimental ECD spectrum of compound **2** exhibiting positive Cotton effect at 241 nm and negative Cotton effect at 320 nm matched well with the (1*R*, 10*S*) isomer and mirrored with (1*S*, 10*R*) isomer in the calculated ECD spectra (Figure 4), permitting assignment of the (1*R*, 10*S*) absolute configuration for compound **2**.

Molestin B (**3**) was obtained as a colorless oil having a molecular formula of C_15_H_24_O_2_ as deduced from the protonated molecular [M + H]^+^ at *m*/*z* 237.1853 (calcd for C_15_H_25_O_2_, 237.1849) in the (+)-HRESIMS spectrum and ^13^C NMR data (Table 1). The ^13^C NMR and DEPT spectra showed 15 carbon signals, which were assigned as four methyls, four methylenes, three methines, and four quaternary carbons (including two olefinic, one oxygenated, and one conjugated carbonyl). Additionally, the characteristic carbon resonances at *δ*_C_ 174.0 (C-5), 138.6 (C-4), and 208.7 (C-3) attributed to an *α*,*β*-unsaturated ketone group were also observed. Considering all these data as well as four degrees of unsaturation, compound **3** was predicted to be a bicyclic sesquiterpene. The presence of an isopropyl group was verified by the ^1^H–^1^H COSY correlations of H_3_-13/H-12/H_3_-14 (Figure 2). The aforementioned information together with the sequential ^1^H–^1^H COSY correlations of H_2_-8/H_2_-9/H-10/H-1/H_2_-2 implied that the structure of **3** was analogous to the guaiane-type sesquiterpene torilolone [24]. The HMBC correlations from H_3_-11 to C-3, C-4, and C-5; from H_3_-13 and H_3_-14 to C-7; from H_3_-15 to C-1, C-9, and C-10; from H_2_-2 to C-1, C-3, C-5, and C-10; and from H_2_-6 to C-4, C-5, C-7, and C-8 further constructed the planar structure of compound **3** (Figure 2). The NOESY correlations of H-10/H_3_-13 suggested the cofacial arrangement of H_3_-15 and 7-OH. In addition, the *cis*-disposition of H-1 and H-10 was deduced from chemical shift of H-10 (*δ*_H_ 2.08) and H_3_-15 (*δ*_H_ 0.65) in ^1^H NMR spectrum recorded in CDCl_3_ [5,25]. The overall pattern of experimental ECD spectrum of compound **3** matched well with the calculated ECD curve for (1*S*, 7*R*, 10*S*) isomer (Figure 4), demonstrating the (1*S*, 7*R*, 10*S*) absolute configuration for compound **3**.

Molestin C (**4**), a yellow oil, had a molecular formula of C_15_H_22_O_2_ based on its (+)-HRESIMS ion at *m*/*z* 235.1693 [M + H]^+^ (calcd for C_15_H_23_O_2_, 235.1693), requiring five degrees of unsaturation. Compound **4** shared a similar guaiane-type sesquiterpene skeleton with compound **3** by comparison of their 1D NMR data (Table 1 and Table 2). The spin-spin coupling system of H_2_-8/H_2_-9/H-10/H_3_-15 and H_3_-13/H-12/H_3_-14 deduced from ^1^H–^1^H COSY spectrum and HMBC correlations from H_3_-11 to C-3, C-4 and C-5; from H-12 to C-6 and C-8; from H_3_-15 to C-1, C-9 and C-10; from H_2_-3 to C-1 and C-5; from H-6 to C-1, C-4, and C-8; and from H-10 to C-2 and C-5 (Figure 2) completed the planner structure assignment as depicted for compound **4**. The *E*-geometry of the Δ^6(7)^-double bond was verified by NOESY correlations of H-6/H_3_-13 and H-6/H_3_-14. To determine the absolute configuration of compound **4**, ECD spectra of the four configurational isomers were calculated. Results revealed that the experimental ECD spectrum of compound **4** was consistent with the calculated ECD spectrum for (4*R*, 10*R*) configuration only (Figure 4). Accordingly, the (4*R*, 10*R*) absolute configuration was assigned for compound **4**.

Molestin D (**5**) was isolated as a yellow oil, with a molecular formula of C_13_H_20_O_3_ based on (+)-HRESIMS ion at *m*/*z* 242.1753 [M + NH_4_]^+^ (calcd for C_13_H_24_O_3_N, 242.1751) and ^13^C NMR data (Table 1), implying four degrees of unsaturation. Typical resonances at *δ*_C_162.7 (C-5), 127.1 (C-6), and 205.3 (C-7) in ^13^C NMR spectrum indicated the presence of an *α*,*β*-unsaturated ketone group. The comparisons of 1D NMR spectroscopic data of compound **5** with those of **1**–**4** (Table 1 and Table 2) indicated that compound **5** was likely to be a de-isopropyl guaiane-type sesquiterpene. That speculation was confirmed by ^1^H–^1^H COSY correlations of H_2_-8/H_2_-9/H-10/H_3_-12 and HMBC correlations from H_2_-2 to C-4, C-5, and C-10; from H-6 to C-1, C-4, and C-8; from H_3_-11 to C-3, C-4, and C-5; and from H_3_-12 to C-1, C-9, and C-10 (Figure 2). Additionally, HMBC correlation from 1-OCH_3_ to C-1 demonstrated the location of the methoxyl at C-1 in compound **5**. NOESY correlation of H-6/H_3_-11 suggested *E*-geometry of the Δ^5^-double bond (Figure 3). Besides, NOESY correlations of 1-OCH_3_/H-10 and 1-OMe/H_3_-11, along with the absence of 1-OCH_3_/H_3_-12 suggested that 1-OCH_3_, H-10, and H_3_-11 were on the same side of the ring system. On the other hand, the *cis*-oritation of 1-OCH_3_ and H-10 was indicated by the chemical shift of H-10 (*δ*_H_ 2.30) in compound **5**, because it was reported that the chemical shifts for *trans*-oritation of 1-OH and H-10 in its analogs are more upfield [26,27]. The (1*R*, 4*S*, 10*S*) absolute configuration for molestin D (**5**) was identified through ECD calculations, in which the experimental ECD spectrum of compound **5** showed good agreement with the calculated ECD spectrum for (1*R*, 4*S*, 10*S*) configuration (Figure 4). 

*ent*-sinulolide C ((+)-**9**), a colorless oil, was deduced to have a molecular formula of C_15_H_24_O_4_ according to the (+)-HRESIMS ions at *m*/*z* 269.1747 [M + H]^+^ (calcd for C_15_H_25_O_4_, 269.1747) and *m*/*z* 291.1565 [M + Na]^+^ (calcd for C_15_H_24_O_4_Na, 291.1567). The ^1^H NMR spectrum (see Section 3.3) exhibited one methoxyl at *δ*_H_ 3.77 (3H, s, 7-OCH_3_), two olefinic singlet methyls at *δ*_H_ 2.02 (3H, s, H_3_-14) and 1.73 (3H, s, H_3_-13), and one terminal triplet methyl at *δ*_H_ 0.83 (3H, t, *J* = 6.7 Hz, H_3_-12). The characteristic carbon signals at *δ*_C_ 168.8 (C-3), 135.7 (C-2), and 202.9 (C-1) in ^13^C NMR spectrum (see Section 3.3) indicated the presence of an *α*,*β*-unsaturated ketone group in compound **9**. Besides, the consecutive correlations of H_2_-8/H_2_-9/H_2_-10/H_2_-11/H_3_-12 observed in ^1^H–^1^H COSY spectrum formed a pentane linear side chain in **9** (Figure 2). The aforementioned data bear remarkable similarities to those of sinulolide C [11]. In addition, further comparative analyses of 1D and 2D NMR data of compound **9** and sinulolide C suggested that the two compounds shared an identical planner structure. NOESY correlation of H-6b (*δ*_H_ 2.38)/H-8b (*δ*_H_ 1.53) in compound **9** clarified the same relative configuration as sinulolide C. However, just as the natural occurring analogs (±)-foedanolide ((±)-**12**) [28], compound **9** was optically inactive, indicating **9** to be a racemate. The chiral HPLC separation of **9** yielded optical pure compounds (+)-**9** and (−)-**9** (sinulolide C), with a ratio of approximately 1:1 (Appendix A). The opposite optical rotations of respective +17.8 (*c* 0.05, MeOH) and −23.5 (*c* 0.05, MeOH), as well as mirror-image-like experimental ECD curves (Figure 4) of compounds (+)-**9** and (−)-**9**, suggested the (4*S*, 5*S*) absolute configuration for *ent*-sinulolide C ((+)-**9**), by comparison with those of sinulolide C, sinularone B [12], and (±)-sinularone J [8]. 

*ent*-sinulolide D ((+)-**10**) was obtained as a colorless oil. Its molecular formula of C_14_H_22_O_4_ was determined by the (−)-HRESIMS ion at *m*/*z* 253.1440 [M − H]^−^ (calcd for C_14_H_21_O_4_, 253.1445). The 1D NMR data of **10** (see Section 3.3) revealed that its structural features were very close to those of **9**, differing only in the absence of a methoxyl at C-7 in **10**. The planar structure of compound **10** was unambiguously proved to be the same with sinulolide D by ^1^H–^1^H COSY and HMBC correlations [11]. Moreover, NOESY correlation of H-6b (*δ*_H_ 2.64)/H-8b (*δ*_H_ 1.79) in compound **10** disclosed the same relative configuration as sinulolide D. Similarly, Compound **10** was initially isolated as a racemic mixture, and was separated by chiral HPLC to afford (+)-**10** ([*α*${]}_{D}^{20}$ +44.6, *c* 0.2, MeOH) and (–)-**10** (sinulolide D, [*α*${]}_{D}^{20}$ −42.1, *c* 0.2, MeOH) with a ratio of 1:1 (Appendix A). Comparing the optical rotations as well as experimental ECD spectra of compounds (+)-**10** and (−)-**10** (Figure 4) with those of analogs [8,11,12], the absolute configuration for *ent*-sinulolide D ((+)-**10**) was clearly assigned as (4*S*, 5*S*). The co-occurrence of compounds **9** and **10** in the extract and the fact that MeOH was used during the extraction process suggested that compound **9** is likely a 7-*O*-methyl artifact of **10**.

*ent*-sinulolide F ((+)-**11**), obtained as a colorless oil, was defined a molecular formula of C_15_H_24_O_5_ by the (+)-HRESIMS ion at *m*/*z* 307.1511 [M + Na]^+^ (calcd for C_15_H_24_O_5_Na, 307.1516). The 1D NMR spectroscopic data of compound **11** (see Section 3.3) were very close to those of compound **9**, except for the presence of a methoxyl (*δ*_H_ 3.39, *δ*_C_ 60.4) and an oxygenated methane (*δ*_H_ 3.60, *δ*_C_ 80.9), instead of one methylene (*δ*_H_ 1.77 and 1.53, *δ*_C_ 37.1) in **9**. The ^1^H–^1^H COSY and HMBC data finally defined the planner structure of compound **11**, which was identical to sinulolide F [11]. The relative configuration of **11** was shown to be the same as that of sinulolide F based on equivalent NMR data and NOESY correlation of H-6a (*δ*_H_ 3.06)/H-8 (*δ*_H_ 3.60). Compound **11** was also separated by chiral HPLC to obtain (+)-**11** and (–)-**11** (sinulolide F), with an enantiomeric ratio of 1:1 (Appendix A). Comparison of the experimental ECD spectra for compounds (±)-**11** (Figure 4) and sinulolide F suggested the (4*S*, 5*S*) configuration for *ent*-sinulolide F ((+)-**11**). However, the configurations at C-8 in compounds (±)-**11** were still undetermined.

*ent*-sinulolide H ((+)-**13**), a colorless oil, had a molecular formula of C_11_H_16_O_5_ based on the (+)-HRESIMS ion at *m*/*z* 251.0895 [M + Na]^+^ (calcd for C_11_H_16_O_5_Na, 251.0890), implying four degrees of unsaturation. In the ^13^C NMR spectrum, three quaternary signals at *δ*_C_ 155.9 (C-3), 127.8 (C-2), and 171.4 (C-1) attributed to an *α*,*β*-unsaturated ester group were exhibited (see Section 3.3). The ^1^H NMR data showed two downfield singlet methyls at *δ*_H_ 1.88 (3H, s, H_3_-9) and 1.85 (3H, s, H_3_-8), which were attached to C-2 and C-3 by HMBC correlations from H_3_-8 to C-1 and C-3, and from H_3_-9 to C-2 and C-4, respectively. The presence of a methyl propionate fragment was confirmed by the ^1^H–^1^H COSY correlation of H_2_-5/H_2_-6 and HMBC correlations from H_2_-5 and 7-OCH_3_ to C-7 (Figure 2). In addition, resonances of one notable downfield quaternary carbon at *δ*_C_ 109.0 (C-4) as well as a methoxyl at *δ*_C_ 50.4 (4-OCH_3_) were observed in the ^13^C NMR spectrum. The above information strongly indicated that compound **13** was likely to have the same planner structure as the 2,3-dimethyl butenolide derivative sinulolide H [11], which was then confirmed by ^1^H–^1^H COSY and HMBC correlations (Figure 2). Compound **13** was further purified by chiral HPLC chromatography to obtain (+)-**13** ([*α*${]}_{D}^{20}$ +8.6, *c* 0.01, MeOH) and (–)-**13** (sinulolide H, [*α*${]}_{D}^{20}$ –10.3, *c* 0.01, MeOH) (Appendix A). The absolute configuration of *ent*-sinulolide H ((+)-**13**) was determined as 4*S* by comparing the optical rotations with those of the analogs sinulolide H ([*α*${]}_{D}^{20}$ –3.2, *c* 0.03, MeOH) and sinularone H ([*α*${]}_{D}^{20}$ +3.7, *c* 0.12, MeOH) [12]. 

Molestin E (**14**) was isolated as a yellow oil and had a molecular formula of C_23_H_28_O_9_ as determined by (+)-HRESIMS ion at *m*/*z* 466.2063 [M + NH_4_]^+^ (calcd for C_23_H_32_O_9_N, 466.2072). Absorption for hydroxy and carbonyl groups at 3370, 1751, and 1698 cm^–1^ were observed in the IR spectrum. The 1D NMR data of compound **14** (Table 3) were similar to those of the co-occurring compounds **15** and **16**. ^1^H NMR spectrum of **14** displayed signals at *δ*_H_ 6.75 (1H, s, H-5), 6.10 (1H, s, H-11), and 4.81 (2H, d, *J* = 6.1 Hz, H_2_-16) ascribed to a trisubstituted furan, a disubstituted butyrolactone ring, and a terminal olefin, respectively, as found in sinulacembranolide A isolated from the soft coral *S. gaweli* [29]. The ^13^C NMR and DEPT spectra of **14** showed the presence of 23 carbon signals which were assigned to four methyls, four methylenes (including one terminal olefinic), six methines (including two oxygenated, two olefinic), and nine quaternary carbons (including one oxygenated, four olefinic, and three carbonyl), indicating compound **14** could be the deacetyl analog of sinulacembranolide A. Connectivity information obtained from 2D NMR, especially ^1^H–^1^H COSY and HMBC experiments, unambiguously confirmed the above speculation, and determined the planar structure of **14** (Figure 2). The relative configuration of **14** was elucidated by NOESY spectrum. NOESY correlation of H-1/H-13 suggested that H-1 and H-13 were on the same side of the ring system. NOESY correlations of H-1/H-2*α* (*δ*_H_ 2.75) and H-7/H-2*α* assigned the *α*-orientation of H-1, H-7, and H-13 (Figure 3). The same side of H_3_-19 and H-10 were determined based on NOESY correlations of H_3_-19/H-10, and their *β*-orientation were finally defined by NOESY correlations of H_3_-19/H-9*β* (*δ*_H_ 2.60), H-10/H-9*β*, H-7/H-9*α* (*δ*_H_ 1.80), and H-13/H-9*α*. That allowed the assignment of the relative configuration of **14** as (1*S**, 7*R**, 8*S**, 10*R**, 13*R**).

The known compounds, including three sesquiterpenes (**6**–**8**), five cyclopentenone derivatives ((–)-**9**–(–)-**11**, and (±)-**12**), one butenolide derivative ((–)-**13**), and five diterpenes (**15**–**19**) were identified as gibberodione [5], polydactin A [30], (5′*Z*)-5-(2′,6′-Dimethylocta-5′,7′-dienyl)-furan-3- carboxylic acid [31], sinulolide C [11], sinulolide D [11], sinulolide F [11], (±)-foedanolide ((±)-sinularone D) [12,28], sinulolide H [11], leptodiol [32], (–)-leptodiol acetate [29], 5-*epi*-sinuleptolide [33], sinuleptolide [34,35], and scabrolide A [36], by comparing their spectroscopic data with those reported earlier.

Considering the reported cytotoxic activities of some sesquiterpenes [30], cyclopentenone derivatives [28], and cembranoid diterpenes [36], all the compounds were evaluated for their cytotoxic activities against HeLa (human cervical carcinoma), HCT-116 (human colon carcinoma), BEL-7402 (human hepatocellular carcinoma), K562 (human leukemia), and Jurkat (human acute leukemia T) tumor cell lines. Only three cembranoid diterpenes showed valuable cytotoxicities against the selected cell lines (IC_50_ < 10 μM). Molestin E (**14**) exhibited cytotoxicities against HeLa and HCT-116 cell lines with IC_50_ values of 5.26 and 8.37 μM, respectively. Compounds **17** and **18** showed cytotoxicities against HeLa cell lines with IC_50_ values of 6.66 and 6.10 μM, respectively. 

Considering some guaiane-type sesquiterpenes and furanoterpenoids previously reported showed inhibitory activities against protein tyrosine phosphatase 1B (PTP1B) [37,38,39], a major negative regulator in insulin signaling pathways [40]. Inhibiting PTP1B activity could increase insulin sensitivity and is expected to be a potential promising therapeutic for type 2 diabetes and obesity [41]. These isolated sesquiterpenes were also assessed for their inhibitory activities against PTP1B. In addition, the results revealed that two guaiane-type sesquiterpenes (**4** and **5**) and a furanosesquiterpene (**8**) displayed strong inhibitory activities against PTP1B with IC_50_ values of 218, 344, and 1.24 μM, respectively, lower than the positive control (the IC_50_ value of sodium orthovanadate was 881 μM). 

## 3. Materials and Methods

### 3.1. General Methods

Optical rotations were measured on a JASCO P-1020 digital polarimeter. Ultra-violet (UV) spectra were measured on a Beckman DU640 spectrophotometer. ECD spectra were obtained on a JASCO J-810 spectropolarimeter. IR spectra were recorded on a Nicolet Nexus 470 FT-IR spectrophotometer in KBr discs. 1D and 2D NMR spectra were recorded on a JEOL JNMECP 500 spectrometer (500 MHz for ^1^H and 125 MHz for ^13^C) using tetramethylsilane as an internal standard. Chemical shifts are expressed in *δ* (ppm) referring to the solvent peaks at *δ*_H_ 7.26 and *δ*_C_ 77.16 for CDCl_3_, and *δ*_H_ 3.31 and *δ*_C_ 49.50 for methanol-*d*_4_ and coupling constant *J* in Hz. HRESIMS data were acquired on a Thermo Scientific LTQ Orbitrap XL mass spectrometer or a Micromass Q-Tof Ultima GLOBAL GAA076 mass spectrometer, equipped with an electrospray ionization (ESI) source, and the ionization mode was positive or negative. Semipreparative HPLC separations were carried out using an Agilent 1100 series instrument with a diode array detector (DAD) detector, equipped with a reversed-phase column (YMC-Pack ODS-A, 5 μm, 250 × 10 mm). Chiral HPLC analysis and resolution were conducted on a chiral analytical column (Daicel Chiralpack IC, 5 μm, 250 × 4.6 mm). Silica gel (200–300 mesh, 300–400 mesh; Qingdao Marine Chemical Co. Ltd., Qingdao, China), ODS silica gel (50 μm, Merck, Darmstadt, Germany) and Sephadex LH-20 (GE Healthcare Bio-Sciences AB, Uppsala, Sweden) were used to perform column chromatography (CC). Precoated silica gel plates (GF254, Qingdao Marine Chemical Co. Ltd., Qingdao, China) were used for thin layer chromatography (TLC) analyses, and spots were visualized under UV light and by spraying with 10% H_2_SO_4_ in EtOH.

### 3.2. Animal Material

The soft coral *S.* cf. *molesta* was collected from the Paracel Islands of the South China Sea in October 2012, at a depth of about 10 m, and was frozen immediately until it was examined. The specimen was identified by Dr. Leen van Ofwegen, a co-author of this paper.. A voucher specimen (NO. XS-2012-22) was deposited in the school of Medicine and Pharmacy, Ocean University of China, China.

### 3.3. Extraction and Isolation

The frozen organism (5.0 kg, wet weight; 2.2 kg, dry weight) was homogenized and extracted with MeOH four times (each time, 5 L, 3 d) at room temperature. The combined solutions were concentrated in vacuum and desalted by anhydrous MeOH three times (0.6 L, 0.5 L, 0.4 L) to yield a residue (95.0 g). The crude extract was subjected to silica gel vacuum liquid chromatography (VLC), eluting with a gradient of petroleum ether (PE)/acetone (150:1–1:1) and subsequently CH_2_Cl_2_/MeOH (20:1–1:1) to obtain seven fractions (Frs. 1–7). Fr. 2 was subjected to silica gel CC using PE/acetone (100:1–5:1 gradient) as eluent to give four subfractions (Frs. 2.1–2.4). Fr. 2.2 was separated by ODS CC (MeOH/H_2_O, 60:40−100:0 gradient) to yield four subfractions (Frs. 2.2.1–2.2.4). Fr. 2.2.2 was further purified by semipreparative HPLC (ODS; MeOH/H_2_O, 60:40) to afford compounds 1 (3.5 mg) and 2 (1.7 mg). Fr. 2.2.3 was purified by semipreparative HPLC (ODS; MeOH/H_2_O, 40:60) to afford compounds 6 (10.0 mg) and 7 (2.1 mg). Fr. 3 was applied to silica gel CC (PE/acetone, 50:1–1:1 gradient) to yield six subfractions (Frs. 3.1–3.6). Fr. 3.3 was further separated by CC (Sephadex LH-20; CHCl_3_/MeOH, 1:1), and then semipreparative HPLC (ODS; MeOH/H_2_O, 40:60) to afford compound 13 (2.5 mg). Fr. 3.5 was purified by ODS CC (MeOH/H_2_O, 40:60−80:20 gradient) and then semipreparative HPLC (ODS; MeOH/H_2_O, 40:60) to yield compounds 3 (32.0 mg), 4 (2.1 mg), and 5 (2.5 mg). Fr. 4 was fractionated into six subfractions (Fr. 4.1−4.6) by silica gel CC (PE/acetone, 40:1–1:1 gradient). Fr. 4.2 was fractionated by ODS CC, eluted with MeOH/H_2_O (30:70−80:20 gradient), followed by semipreparative HPLC (MeOH/H_2_O, 55:45), to afford compounds 10 (5.0 mg) and 12 (18.0 mg). Repeated Sephadex LH-20 CC (CHCl_3_/MeOH, 1:1), followed by ODS CC (MeOH/H_2_O, 30:70−80:20 gradient) and then semipreparative HPLC (MeOH/H_2_O, 45:55), gave compounds 14 (14.0 mg), 15 (33.0 mg), 16 (1.8 mg), 17 (55.0 mg), 18 (43.0 mg), and 19 (1.5 mg) from Fr. 4.5. Fr. 5 was separated by silica gel CC (PE/acetone, 20:1−1:1 gradient) to provide five subfractions (Fr. 5.1−5.5). Fr. 5.5 was separated by Sephadex LH-20 CC (CH_2_Cl_2_/MeOH, 1:1), followed by semipreparative HPLC (MeOH/H_2_O, 55:45), to yield compounds 8 (4.5 mg), 9 (1.6 mg), and 11 (1.5 mg). Chiral separations of (±)-9−(±)-13 (chiral HPLC column; *n*-hexane/*i*-PrOH 80:20, 90:10, 70:30, 97:3, and 85:15, respectively) to resolved into optically pure compounds (+)-9 (0.8 mg), (–)-9 (0.8 mg), (+)-10 (2.1 mg), (−)-10 (2.2 mg), (+)-11 (0.7 mg), (−)-11 (0.7 mg), (+)-12 (8.7 mg), (−)-12 (8.8 mg), (+)-13 (1.2 mg), and (−)-13 (1.2 mg). During the HPLC separations, the injection volume was less than 40 μL (semipreparative HPLC) or 5 μL (chiral HPLC) and the column temperature was maintained at about 30 °C.

*Molestin A* (**1**): colorless oil; [*α*${]}_{D}^{20}$ +10.0 (*c* 0.1, MeOH); ECD (MeOH) *λ*_max_ (Δ*ε*) 241 (−9.1), 328 (3.9) nm; IR (KBr) *ν*_max_ 3151, 2975, 2932, 2872, 1712, 1649,1453, 1379, 1217, 1096 cm^−1^; ^1^H and ^13^C NMR data, see Table 1 and Table 2; HRESIMS *m*/*z* 253.1803 [M + H]^+^ (calcd for C_15_H_25_O_3_, 253.1798).

*Epi-gibberodione* (**2**): colorless oil; [*α*${]}_{D}^{20}$ +9.8 (*c* 0.1, MeOH); ECD (MeOH) *λ*_max_ (Δ*ε*) 242 (10.1), 320 (−5.2) nm; IR (KBr) *ν*_max_ 2954, 2803, 1703, 1652, 1465, 1382, 1190 cm^−1^; ^1^H NMR (500 MHz, CDCl_3_) *δ* 5.84 (1H, br d, *J* = 2.1 Hz, H-6), 2.11 (3H, s, H_3_-11), 1.10 (3H, d, *J* = 6.2 Hz, H_3_-15), 1.07 (3H, d, *J* = 6.6 Hz, H_3_-13), 1.06 (3H, d, *J* = 6.8 Hz, H_3_-14). ^1^H and ^13^C NMR data (methanol-*d*_4_), see Table 1 and Table 2; HRESIMS *m*/*z* 237.1852 [M + H]^+^ (calcd for C_15_H_25_O_2_, 237.1849).

*Molestin B* (**3**): colorless oil; [*α*${]}_{D}^{20}$ +37.9 (*c* 0.1, MeOH); ECD (MeOH) *λ*_max_ (Δ*ε*) 242 (9.9), 316 (−1.4) nm; IR (KBr) *ν*_max_ 3451, 2972, 2850, 1685, 1496, 1383, 1167, 996 cm^−1^; ^1^H and ^13^C NMR data, see Table 1 and Table 2; HRESIMS *m*/*z* 237.1853 [M + H]^+^ (calcd for C_15_H_25_O_2_, 237.1849).

*Molestin C* (**4**): yellow oil; [*α*${]}_{D}^{20}$ +110.9 (*c* 0.1, MeOH); ECD (MeOH) *λ*_max_ (Δ*ε*) 213 (−0.2), 229 (3.3), 298 (5.6), 337 (−10.3) nm; IR (KBr) *ν*_max_ 3450, 2961, 2780, 1683,1506, 1381, 1257,1035 cm^−1^; ^1^H and ^13^C NMR data, see Table 1 and Table 2; HRESIMS *m*/*z* 235.1693 [M + H]^+^ (calcd for C_15_H_23_O_2_, 235.1693).

*Molestin D* (**5**): yellow oil; [*α*${]}_{D}^{20}$ +39.4 (*c* 0.2, MeOH); ECD (MeOH) *λ*_max_ (Δ*ε*) 212 (22.8), 238 (−44.2), 322 (10.9) nm; IR (KBr) *ν*_max_ 3417, 2966, 1666, 1458, 1052, 916, 870 cm^−1^; ^1^H and ^13^C NMR data, see Table 1 and Table 2; HRESIMS *m*/*z* 242.1753 [M + NH_4_]^+^ (calcd for C_13_H_24_O_3_N, 242.1751).

*ent-sinulolide C* ((+)**-9**): colorless oil; [*α*${]}_{D}^{20}$ +17.8 (*c* 0.05, MeOH); ECD (MeOH) *λ*_max_ (Δ*ε*) 210 (40.9), 236 (−79.5), 314 (22.1) nm; UV (MeOH) *λ*_max_ (log *ε*) 224 (2.16) nm; IR (KBr) *ν*_max_ 3445, 2985, 2921, 1730, 1643, 1457, 1370, 1163 cm^−1^; ^1^H NMR (500 MHz, CDCl_3_) *δ* 2.85 (1H, dd, *J* = 11.7, 2.9 Hz, H-5), 3.15 (1H, dd, *J* = 18.5, 3.2 Hz, H-6a), 2.38 (1H, dd, *J* = 18.6, 11.8 Hz, H-6b), 1.77 (1H, td, *J* = 13.5, 5.3 Hz, H-8a), 1.53 (1H, td, *J* = 12.1, 3.8 Hz, H-8b), 0.76 (1H, m, H-9a), 0.64 (1H, m, H-9b), 1.20 (4H, m, H_2_-10, H_2_-11), 0.83 (3H, t, *J* = 6.7 Hz, H_3_-12), 1.73 (3H, s, H_3_-13), 2.02 (3H, s, H_3_-14), 3.77 (3H, s, 7-OCH_3_); ^13^C NMR (125 MHz, CDCl_3_) *δ* 202.9 (C, C-1), 135.7 (C, C-2), 168.8 (C, C-3), 80.6 (C, C-4), 55.2 (CH, C-5), 30.0 (CH_2_, C-6), 175.4 (C, C-7), 37.1 (CH_2_, C-8), 25.2 (CH_2_, C-9), 32.1 (CH_2_, C-10), 22.5 (CH_2_, C-11), 14.1 (CH_3_, C-12), 8.0 (CH_3_, C-13), 11.7 (CH_3_, C-14), 52.6 (CH_3_, 7-OCH_3_); HRESIMS *m*/*z* 269.1747 [M + H]^+^ (calcd for C_15_H_25_O_4_, 269.1747), *m*/*z* 291.1565 [M + Na]^+^ (calcd for C_15_H_24_O_4_Na, 291.1567).

*ent-sinulolide D* ((+)-**10**): colorless oil; [*α*${]}_{D}^{20}$ +44.6 (*c* 0.2, MeOH); UV (MeOH) *λ*_max_ (log *ε*) 228 (2.60) nm; ECD (MeOH) *λ*_max_ (Δ*ε*) 229 (−89.1), 320 (9.0) nm; IR (KBr) *ν*_max_ 3422, 2959, 2860, 1735, 1610, 1469, 1365, 1103 cm^−1^; ^1^H NMR (500 MHz, CDCl_3_) *δ* 2.94 (1H, overlap, H-5), 2.96 (1H, overlap, H-6a), 2.64 (1H, d, *J* = 15.5 Hz, H-6b), 2.01 (1H, td, *J* = 14.4, 4.2 Hz, H-8a), 1.79 (1H, m, H-8b), 135−1.25 (5H, m, H-9a, H_2_-10, and H_2_-11), 1.11 (1H, m, H-9b), 0.89 (3H, t, *J* = 6.8 Hz, H_3_-12), 1.76 (3H, s, H_3_-13), 2.07 (3H, s, H_3_-14); ^13^C NMR (125 MHz, CDCl_3_) *δ* 204.7 (C, C-1), 139.2 (C, C-2), 167.2 (C, C-3), 92.4 (C, C-4), 46.7 (CH, C-5), 32.6 (CH_2_, C-6), 174.5 (C, C-7), 34.7 (CH_2_, C-8), 23.4 (CH_2_, C-9), 31.9 (CH_2_, C-10), 22.6 (CH_2_, C-11), 14.1 (CH_3_, C-12), 8.3 (CH_3_, C-13), 12.4 (CH_3_, C-14); HRESIMS *m*/*z* 253.1440 [M − H]^+^ (calcd for C_14_H_21_O_4_, 253.1445).

*ent-sinulolide F* ((+)-**11**): colorless oil; [*α*${]}_{D}^{20}$ +9.8 (*c* 0.04, MeOH); UV (MeOH) *λ*_max_ (log *ε*) 229 (2.93) nm; ECD (MeOH) *λ*_max_ (Δ*ε*) 208 (90.9), 226 (−28.5), 246 (26.9) nm; IR (KBr) *ν*_max_ 3385, 2927, 1722, 1635, 1480, 1377, 1145 cm^−1^; ^1^H NMR (500 MHz, CDCl_3_) *δ* 2.91 (1H, dd, *J* = 19.1, 12.5 Hz, H-5), 3.06 (1H, dd, *J* = 12.5, 4.6 Hz, H-6a), 2.59 (1H, dd, *J* = 19.1, 4.6 Hz, H-6b), 3.60 (1H, dd, *J* = 8.8, 2.9 Hz, H-8), 1.52 (1H, m, H-9a), 1.34 (5H, m, H-9b, H_2_-10, H_2_-11), 0.92 (3H, t, *J* = 7.1 Hz, H_3_-12), 1.78 (3H, s, H_3_-13), 2.13 (3H, s, H_3_-14), 3.39 (3H, s, 8-OCH_3_); ^13^C NMR (500 MHz, CDCl_3_) *δ* 204.6 (C, C-1), 140.1 (C, C-2), 165.8 (C, C-3), 93.0 (C, C-4), 44.2 (CH, C-5), 32.3 (CH_2_, C-6), 174.4 (C, C-7), 80.9 (CH, C-8), 30.7 (CH_2_, C-9), 28.8 (CH_2_, C-10), 23.0 (CH_2_, C-11), 14.2 (CH_3_, C-12), 8.5 (CH_3_, C-13), 13.2 (CH_3_, C-14), 60.4 (CH_3_, 8-OCH_3_); HRESIMS *m*/*z* 307.1511 [M + Na]^+^ (calcd for C_15_H_24_O_5_Na, 307.1516).

*ent-sinulolide H* ((+)-**13**): colorless oil, [*α*${]}_{D}^{20}$ +8.6 (*c* 0.01, MeOH); UV (MeOH) *λ*_max_ (log *ε*) 233 (2.10) nm; IR (KBr) *ν*_max_ 2815, 2700, 1750, 1485, 1311, 1084 cm^−1^; ^1^H NMR (500 MHz, CDCl_3_) *δ* 2.52 (1H, m, H-5a), 2.38 (1H, m, H-5b), 2.30 (1H, m, H-6a), 1.96 (1H, m, H-6b), 1.85 (3H, s, H_3_-8), 1.88 (3H, s, H_3_-9), 3.08 (3H, s, 4-OCH_3_), 3.66 (3H, s, 7-OCH_3_); ^13^C NMR (125 MHz, CDCl_3_) *δ* 171.4 (C, C-1), 127.8 (C, C-2), 155.9 (C, C-3), 109.0 (C, C-4), 28.0 (CH_2_, C-5), 31.4 (CH_2_, C-6), 173.4 (C, C-7), 8.6 (CH_3_, C-8), 10.9 (CH_3_, C-9), 50.4 (CH_3_, 4-OCH_3_), 51.9 (CH_3_, 7-OCH_3_); HRESIMS *m*/*z* 251.0895 [M + Na]^+^ (calcd for C_11_H_16_O_5_Na, 251.0890).

*Molestin E* (**14**): yellow oil; [*α*${]}_{D}^{20}$ +13.2 (*c* 0.2, MeOH); UV (MeOH) *λ*_max_ (log *ε*) 264 (2.96) nm; IR (KBr) *ν*_max_ 3481, 1756, 1721, 1580, 1396, 1185 cm^−1^; ^1^H and ^13^C NMR data, see Table 3; HRESIMS *m*/*z* 466.2063 [M + NH_4_]^+^ (calcd for C_23_H_32_O_9_N, 466.2072).

### 3.4. ECD Calculations of Compounds ***1**–**5***

The quantum chemical calculations were performed by using the density functional theory (DFT) as implemented in Gaussian 09 [42]. The initial structures of compounds 1–5 were built with Spartan 10 software and all trial structures were first minimized based on molecular mechanics calculations. Conformational searches were performed by Spartan 10 software using Merck Molecular Force Field (MMFF), and conformers occurring within a 10 kcal/mol energy window from the global minimum were chosen for geometry optimization in the gas phase with the DFT method at the B3LYP/6-31G (d,p) level. The stable conformations of 1–5 were calculated for ECD spectra using TD-DFT method with the basis set RB3LYP/ DGDZVP [23]. Solvent effects of MeOH were evaluated at the same DFT level by using the SCRF/PCM method. The ECD spectra were combined after Boltzmann weighting according to their population contribution (Appendix A).

### 3.5. Cytotoxicity Assay

In vitro cytotoxicities were determined by MTT (3-(4,5-dimethylthiazol-2-yl)-2,5-diphenyl-2*H*- tetrazolium bromide) colorimetric method [43] against K562 and Jurkat cell lines, and SRB (Sulforhodamine B) method [44] against HeLa, HCT-116, and BEL-7402 cell lines, with adriamycin as a positive control, and compounds with IC_50_ values > 50 μM were considered to be inactive in cytotoxicity assays.

### 3.6. PTP1B Inhibitory Assay

The PTP1B inhibitory activities of the isolates were evaluated by the method of pNPP [45], using sodium orthovanadate as a positive control.

## 4. Conclusions

During the first chemical and biological investigation on the soft coral *S.* cf. *molesta*, ten new compounds (**1**–**5**, (+)-**9**–(+)-**11**, (+)-**13**, and **14**) and 14 known related metabolites (**6**–**8**, (–)-**9**–(–)-**11**, (±)-**12**, (–)-**13**, **15**–**19**) were obtained. Since 1975, only seven guaiane-type sesquiterpenes and two secoguaiane-type sesquiterpenes have been isolated from the soft corals of the genus *Sinularia*, including the Red Sea soft corals *S. gardineri* [46] and *S. terspilli* [47], the Formosan soft corals *S. leptoclados* [48] and *S. gibberosa* [5], the Hainan Islands soft corals *S. numerosa* [49] and *Sinularia* sp. [50], and the South China Sea soft corals *Sinularia* sp. [51,52]. Our present study on the soft coral *S.* cf. *molesta* collected from the Paracel Islands of the South China Sea yielded two new secoguaiane-type sesquiterpenes (**1** and **2**), and three new guaiane-type sesquiterpenes (**3**–**5**), which were distinctly different from the africane-type sesquiterpenes from Moyli Island soft coral *S. intacta* [19]. Compounds (+)-**12** and **15** were first encountered in the genus of *Sinularia.* That enriched the specific chemo-diversities of the genus *Sinularia*. The cytotoxicities of three cembranoid diterpenes (**14**, **17**, and **18**) and PTP1B inhibitory activities of two guaiane-type sesquiterpenes (**4**, **5**) and one furanosesquiterpene (**8**) were unveiled for pharmaceutical potentials. 

## Figures and Tables

**Figure 1 marinedrugs-16-00517-f001:**
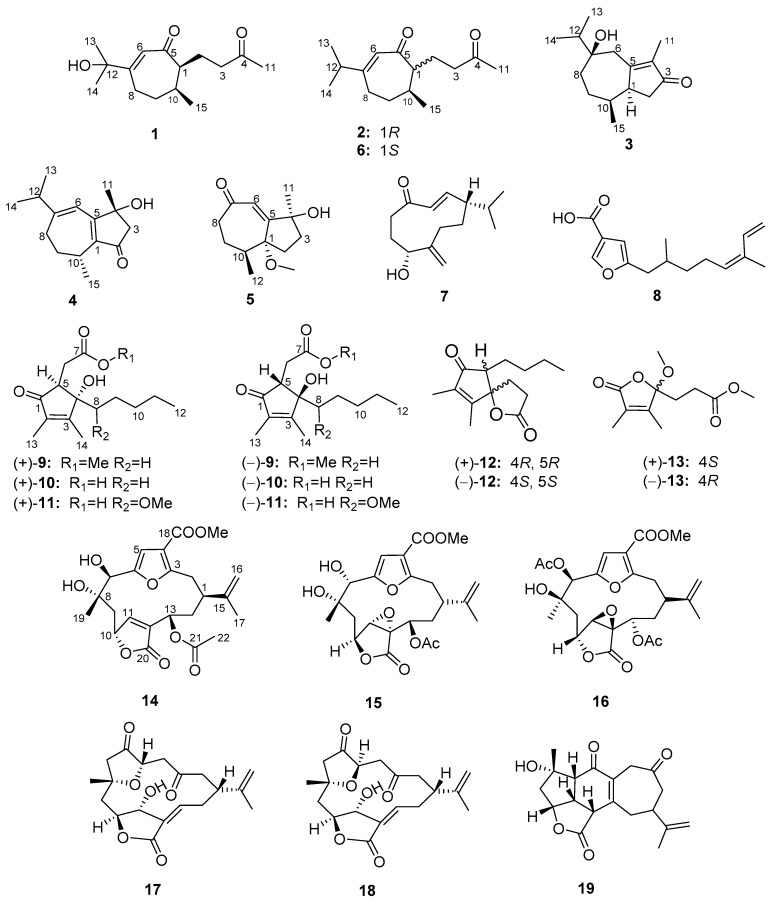
Structures of compounds **1**–**19**.

**Figure 2 marinedrugs-16-00517-f002:**
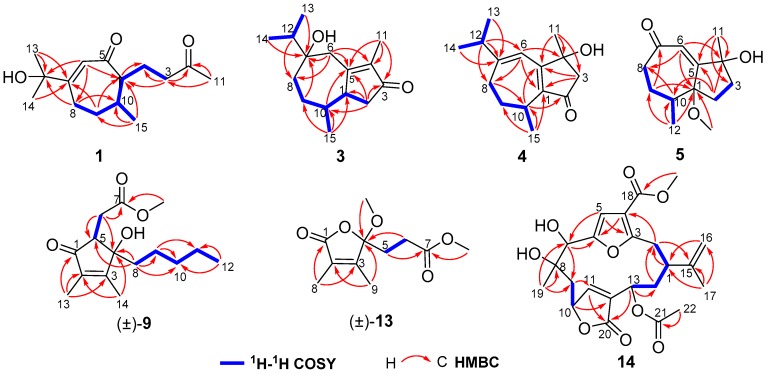
^1^H−^1^H COSY and key HMBC correlations of compounds **1**, **3**, **4**, **5**, (±)-**9**, (±)-**13**, and **14**.

**Figure 3 marinedrugs-16-00517-f003:**
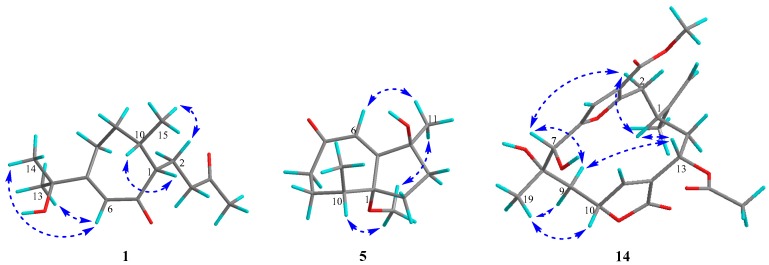
Key NOESY correlations of compounds **1**, **5**, and **14**.

**Figure 4 marinedrugs-16-00517-f004:**
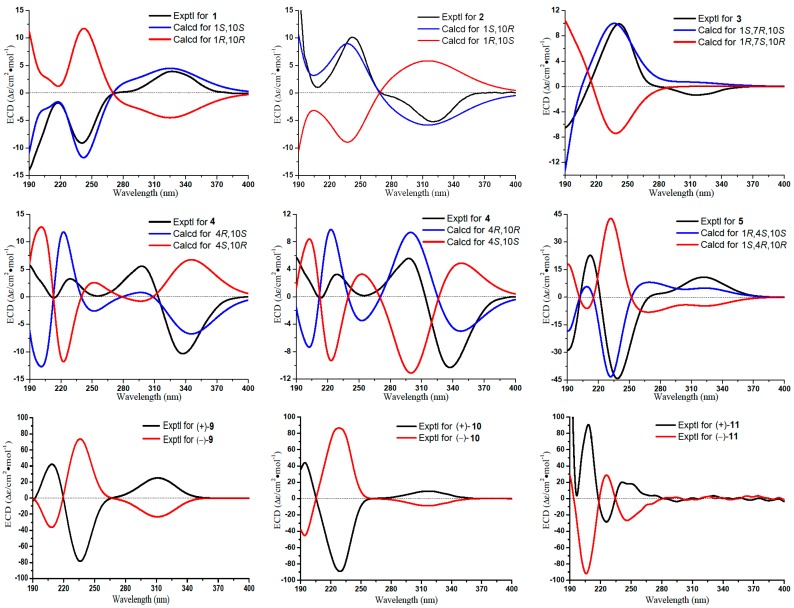
Experimental and calculated ECD spectra of compounds **1**–**5** and experimental ECD spectra of compounds (±)-**9**–(±)-**11**.

**Table 1 marinedrugs-16-00517-t001:** ^13^C NMR data for compounds **1**−**5** (*δ* in ppm).

Position	1 *^a^*	2 *^b^*	3 *^a^*	4 *^a^*	5 *^a^*
1	53.0, CH	59.2, CH	46.1, CH	142.8, C	89.2, C
2	22.3, CH_2_	25.1, CH_2_	41.0, CH_2_	204.2, C	28.9, CH_2_
3	42.0, CH_2_	42.5, CH_2_	208.7, C	51.1, CH_2_	38.3, CH_2_
4	209.1, C	211.4, C	138.6, C	76.3, C	78.7, C
5	203.9, C	208.0, C	174.0, C	164.6, C	162.7, C
6	127.1, CH	127.1, CH	38.9, CH_2_	114.4, CH	127.1, CH
7	168.7, C	170.2, C	75.6, C	165.9, C	205.3, C
8	27.8, CH_2_	29.3, CH_2_	31.7, CH_2_	27.73, CH_2_	40.3, CH_2_
9	36.7, CH_2_	35.9, CH_2_	30.7, CH_2_	29.4, CH_2_	27.6, CH_2_
10	33.9, CH	35.9, CH	34.6, CH	29.1, CH	34.7, CH
11	30.1, CH_3_	29.8, CH_3_	11.6, CH_3_	27.67, CH_3_	26.9, CH_3_
12	74.0, C	38.6, CH	42.1, CH	39.3, CH	16.9, CH_3_
13	28.4, CH_3_	21.5, CH_3_	17.6, CH_3_	21.7, CH_3_	
14	28.6, CH_3_	21.3, CH_3_	17.4, CH_3_	21.3, CH_3_	
15	16.3, CH_3_	20.3, CH_3_	8.3, CH_3_	19.6, CH_3_	
1-OCH_3_					49.6, CH_3_

*^a^* Recorded in CDCl_3_ at 125 MHz; *^b^* Recorded in methanol-*d*_4_ at 125 MHz.

**Table 2 marinedrugs-16-00517-t002:** ^1^H NMR data for compounds **1**−**5** (*δ* in ppm, *J* in Hz).

Position	1 *^a^*	2 *^b^*	3 *^a^*	4 *^a^*	5 *^a^*
1	2.86, m	2.37, overlap	2.08, overlap		
2	1.59, m	1.90, m	2.03, d (19.5)		2.07, m
		1.79, m	2.54, overlap		1.70, overlap
3	2.48, m	2.43, m		2.56, d (5.7)	1.89, m
	2.28, m				1.70, overlap
4					
5					
6	6.22, s	5.83, br d (1.8)	2.73, d (19.6)	6.11, s	6.15, s
			2.54, overlap		
7					
8	2.56, m	2.50, m	1.74, overlap	2.41, dt (11.2, 2.2)	2.77, m
	2.38, m	2.34, overlap		2.37, m	2.50, m
9	2.19, m	1.69, overlap	1.62, m	1.79, m	2.16, m
	1.08, m			1.65, m	1.43, m
10	2.10, m	1.69, overlap	2.08, overlap	2.99, m	2.30, m
11	2.12, s	2.10, s	1.66, s	1.50, s	1.32, s
12		2.36, overlap	1.71, overlap	2.51, m	0.95, d (7.1)
13	1.38, s	1.09, d (6.9)	1.00, d (7.0)	1.12, d (6.9)	
14	1.38, s	1.08, d (6.8)	0.99, d (7.0)	1.12, d (6.9)	
15	0.80, d (6.7)	1.11, d (6.6)	0.65, d (7.1)	1.03, d (7.0)	
1-OCH_3_					3.12, s

*^a^* Recorded in CDCl_3_ at 500 MHz; *^b^* Recorded in methanol-*d*_4_ at 500 MHz.

**Table 3 marinedrugs-16-00517-t003:** ^1^H and ^13^C NMR data for compound **14** (*δ* in ppm, *J* in Hz).

Position	*δ* _H_ *^a^*	*δ* _C_ *^b^*
1	2.30, br t (10.4)	41.5, CH
2	3.58, dd (10.8, 4.4)	32.1, CH_2_
	2.75, d (15.2)	
3		160.4, C
4		115.7, C
5	6.75, s	108.8, CH
6		152.5, C
7	4.53, s	75.9, CH
8		73.7, C
9	2.60, m 1.80, m	42.7, CH_2_
10	4.95, dd (10.8, 4.9)	78.1, CH
11	6.10, s	154.6, CH
12		129.7, C
13	5.51, dd (11.2, 4.9)	66.9, CH
14	2.60, m 1.89, td (13.1, 4.8)	36.2, CH_2_
15		148.6, C
16	4.81, d (6.1)	110.7, CH_2_
17	1.80, s	20.8, CH_3_
18		163.9, C
19	1.41, s	18.8, CH_3_
20		170.1, C
21		170.5, C
22	1.98, s	21.1, CH_3_
18-OCH_3_	3.88, s	52.0, CH_3_

*^a^* Recorded in CDCl_3_ at 500 MHz; *^b^* Recorded in CDCl_3_ at 125 MHz.

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
