# Peer review of "Metabolites from the Paracel Islands Soft Coral Sinularia cf. molesta"

_marinedrugs, 2018, doi:10.3390/md16120517_

Reviewer 1 Report

Cu et al have studied and identified a range of novel metabolites from the soft coral Sinularia cf.molesta, the work is well written and a thorough characterisation of these metabolites has been performed well. A number of details from the materials and methods is missing most notably the NMR and mass spectrometry methods, however with revisions this manuscript would be a good fit for marine drugs and appeal to a wide audience.

Would be useful to define a metabolite at some point if this is the terminology being used for the manuscript.

Introduction:

Page 1, line 37: remove the “s” from invertebrates

Results and discussion:

Page 3, line 87, Change to Ahmed et al.

Page 9, lines 266-272: What was the justification for using these particular cell lines? How is valuable cytotoxicity defined? What are the mechanisms of this cytotoxicity? Will these compounds be studied further to follow up on these findings?

Materials and methods:

Page 9 lines 289-290: What mass analyzer does the Finnigan MAT95 mass spectrometer have? (QTOF, Triple quad e.t.c)

Page 10 Animal material: How many samples were taken were multiple extract done? Or just used 1 animal and assumed they are all the same?

Page 10, 307-334: There needs to be a full description of the LC methods, injection volume, gradient time program, column temperature e.t.c

There needs to be methods for the NMR and mass spectrometry experiments in the manuscript or supplementary.

Supplementary information

For the HRESIMS data there are clearly some fragments present in the spectra. Could these also be annotated with formula and detail the loss from the parent ion as this will further support the other characterisation performed in this work.

Author Response

Point 1: Would be useful to define a metabolite at some point if this is the terminology being used for the manuscript.

Response 1: According to the reviewer' comment, we have added a brief description of metabolite in the revised manuscript (See page 1, lines 38 and 39 in the revised manuscript).

Point 2: Introduction: Page 1, line 37: remove the “s” from invertebrates.

Response 2: The s” has been removed in the revised manuscript (See page 1, line 37 in the revised manuscript).

Point 3: Results and discussion: Page 3, line 87, Change to Ahmed et al.

Response 3: “Atallah F. Ahmed” has been change to “Ahmed et al.” in the revised manuscript (See page 3, line 89 in the revised manuscript).

Point 4: Page 9, lines 266-272: What was the justification for using these particular cell lines? How is valuable cytotoxicity defined? What are the mechanisms of this cytotoxicity? Will these compounds be studied further to follow up on these findings?

Response 4: 1) We have made the explanations for the selection of tumor cell lines in the revised manuscript (See page 9, lines 268 and 269 in the revised manuscript). 2) Compounds with IC50 values<10 μM were considered to have valuable cytotoxicities and compounds with IC50 values >50 μM were considered to be inactive in cytotoxicity assays (See page 12, lines 412 and 413 in the revised manuscript). 3) The mechanisms of the cytotoxicity are not entirely clear, and we will study these active compounds further in the future work.

Point 5: Materials and methods: Page 9 lines 289-290: What mass analyzer does the Finnigan MAT95 mass spectrometer have? (QTOF, Triple quad e.t.c)

Response 5: The “Finnigan MAT95” has been corrected as “Thermo Sientific LTQ Orbitrap XL” in the revised manuscript (See page 9, line 294 in the revised manuscript).

Point 6: Page 10 Animal material: How many samples were taken were multiple extract done? Or just used 1 animal and assumed they are all the same?

Response 6: In the experiment, 2.2 kg (dry weight) of the rare soft coral Sinularia cf. molesta samples were extracted together and the results are the reflection of these samples.

Point 7: Page 10, 307-334: There needs to be a full description of the LC methods, injection volume, gradient time program, column temperature e.t.c

Response 7: Every compound was obtained by repeated and isocratic-elution HPLC separations. The mobile phase ratios used in the experiments have been shown in the revised manuscript (See page 10, lines 320-336 in the revised manuscript). The injection volumes and the column temperature have been added in the revised manuscript, as suggested by the reviewer (See page 10, lines 339-341 in the revised manuscript).

Point 8: There needs to be methods for the NMR and mass spectrometry experiments in the manuscript or supplementary.

Response 8: According to the reviewer' comment, we have added some descriptions of the general method of the NMR and MS experiments (See pages 9 and 10, lines 290-296 in the revised manuscript), and some other experimental parameters are different for different compounds.

Point 9: Supplementary information, For the HRESIMS data there are clearly some fragments present in the spectra. Could these also be annotated with formula and detail the loss from the parent ion as this will further support the other characterisation performed in this work.

Response 9: In the study, NMR spectra interpretation was the most compelling evidence for the structure identification of compounds. Because the HRESIMS data lacked tandem mass information, the fragment ions were not annotated.

Reviewer 2 Report

The paper describes well the isolation and structure elucidation of 10 new and 14 known secondary metabolites from a Sinularia sp.

It’s a pity that the carbon numbering  of compounds is not according to IUPAC.

As for the double bond configuration of compounds 1 and 5 , is a Z configuration at all possible ?

Line 134; what is the distance between H-10 and Me-13 to enable a NOESY correlation ?

Author Response

Point 1: It’s a pity that the carbon numbering of compounds is not according to IUPAC.

Response 1: The carbon numberings of the new compounds were based on those of the previous reported analogues (Ahmed, A. F., J. Nat. Prod. 2005, 68, 1208-1212).  

Point 2: As for the double bond configuration of compounds 1 and 5, is a Z configuration at all possible ?

Response 2: The configurations of the double bonds were determined by NOESY correlation as E geometry.

Point 3: Line 134; what is the distance between H-10 and Me-13 to enable a NOESY correlation ?

Response 3: The distance between H-10 and Me-30 is 2.4 Å, which enable a NOESY correlation.